# 3D Clumps/Extracellular Matrix Complexes of Periodontal Ligament Stem Cells Ameliorate the Attenuating Effects of LPS on Proliferation and Osteogenic Potential

**DOI:** 10.3390/jpm11060528

**Published:** 2021-06-09

**Authors:** Spoorthi Ravi Banavar, Swati Yeshwant Rawal, Shaju Jacob Pulikkotil, Umer Daood, Ian C. Paterson, Fabian Amalraj Davamani, Mikihito Kajiya, Hidemi Kurihara, Suan Phaik Khoo, Eng Lai Tan

**Affiliations:** 1Oral Diagnostic and Surgical Sciences, School of Dentistry, International Medical University, 126, Jalan Jalil Perkasa 19, Bukit Jalil, Kuala Lumpur 57000, Malaysia; suanphaik_khoo@imu.edu.my; 2Department of Surgical Sciences, Marquette University, 1250 W. Wisconsin Ave, Milwaukee, WI 53233, USA; swati.rawal@marquette.edu; 3Clinical Dentistry, School of Dentistry, International Medical University, 126, Jalan Jalil Perkasa 19, Bukit Jalil, Kuala Lumpur 57000, Malaysia; Shaju_Jacob@imu.edu.my (S.J.P.); umerdaood@imu.edu.my (U.D.); 4Department of Oral Craniofacial Sciences, Faculty of Dentistry, University of Malaya, Kuala Lumpur 50603, Malaysia; ipaterson@um.edu.my; 5Oral Cancer Research & Coordinating Centre, Faculty of Dentistry, University of Malaya, Jalan Profesor Diraja Ungku Aziz, Kuala Lumpur 50603, Malaysia; 6School of Health Sciences, International Medical University, Kuala Lumpur 57000, Malaysia; fabian_davamani@imu.edu.my; 7Department of Periodontal Medicine, Graduate School of Biomedical and Health Sciences, Hiroshima University, Hiroshima 734-8553, Japan; mkajiya@Hiroshima-u.ac.jp; 8Dental Academy, 1-6-2 Higashiyanagi, Kudamatsu City 744-0017, Japan; hkuri@hiroshima-u.ac.jp; 9School of Pharmacy, International Medical University, Kuala Lumpur 57000, Malaysia; englai_tan@imu.edu.my

**Keywords:** stem cells, mesenchymal stem cells, periodontal stem cells, 3D cells, organoids, periodontitis, regeneration, osteogenic potential, lipopolysaccharide, proliferation, Alizarin-Red, cetylpyridinium chloride, cytokines

## Abstract

Background: The effects of lipopolysaccharide (LPS) on cell proliferation and osteogenic potential (OP) of MSCs have been frequently studied. Objective: to compare the effects of LPS on periodontal-ligament-derived mesenchymal stem cells (PDLSCs) in monolayer and 3D culture. Methods: The PDLSCs were colorimetrically assessed for proliferation and osteogenic potential (OP) after LPS treatment. The 3D cells were manually prepared by scratching and allowing them to clump up. The clumps (C-MSCs) were treated with LPS and assessed for Adenosine triphosphate (ATP) and OP. Raman spectroscopy was used to analyze calcium salts, DNA, and proline/hydroxyproline. Multiplexed ELISA was performed to assess LPS induced local inflammation. Results: The proliferation of PDLSCs decreased with LPS. On Day 28, LPS-treated cells showed a reduction in their OP. C-MSCs with LPS did not show a decrease in ATP production. Principal bands identified in Raman analysis were the P–O bond at 960 cm^−1^ of the mineral component, 785 cm^−1^, and 855 cm^−1^ showing qualitative changes in OP, proliferation, and proline/hydroxyproline content, respectively. ELISA confirmed increased levels of IL-6 and IL-8 but with the absence of TNF-α and IL-1β secretion. Conclusions: These observations demonstrate that C-MSCs are more resistant to the effects of LPS than cells in monolayer cell culture. Though LPS stimulation of C-MSCs creates an early pro-inflammatory milieu by secreting IL-6 and IL-8, PDLSCs possess inactivated TNF promoter and an ineffective caspase-1 activating process.

## 1. Introduction

Multipotent, self-renewing mesenchymal stromal cells (MSCs) have the potential to differentiate into a variety of cell types. Cells in the periodontal ligament’s perivascular space possess characteristics of mesenchymal stem cells [1]. Like other MSCs, PDLSCs have been cultured and are employed in tissue regeneration experiments since their first isolation in 2004 [2]. Since the early 1900s, two-dimensional (2D) cell cultures have been the preferred method for in vitro studies. However, it is reported that 2D cultures do not accurately represent the cells and tissues [3], simply because the 2D cell culture models cannot mimic the physiology and cellular environments found in vivo [3,4,5]. Multicellular tumor spheroids were first described in the 1970s by culturing tumor cells in a non-adherent environment, resulting in scaffold-free, self-assembled, three-dimensional cellular aggregates [6]. Since then, significant advances in three-dimensional (3D) cell cultures, including organ-on-a-chip technologies, have allowed researchers to review complex aspects of human physiology, pathology, and drug responses in vitro [7].

Few researchers have worked on organoid cell culture in the absence of tissue scaffolds to overcome scaffold’s disadvantages [8,9]. Takewaki et al., in 2017, reported transplantation of bone marrow-derived 3D MSCs into periodontal defects in dogs to successfully induce tissue regeneration without the use of any scaffold [10]. The periodontal ligament (PDL) represents a fibrous network connecting the cementum on the tooth root and the alveolar bone. It serves many functions, such as tooth support, nutrition, and protection [11]. Kotaro et al., in 2020, co-cultured human 3D PDLSCs and vascular endothelial cells and demonstrated enhanced tissue regeneration (TR) capabilities [12].

Periodontitis, a chronic inflammatory disease, is often treated with or without the scaffold and can be refractory to the treatment rendered. It encompasses a continuous phase of inflammation due to inadequate resolution of the initial inflammatory response. One of the important etiological factors for periodontitis is the presence and proliferation of gram-negative organisms and their toxins. Lipopolysaccharide (LPS) is a vital component of the outer membrane of these gram-negative organisms. Lipopolysaccharides remain in the periodontal environment even after removing the organisms by antibiotics, which contributes to the chronic nature of periodontal diseases. Lipopolysaccharides induce the expression of pro-inflammatory cytokines enduring the inflammatory process and decrease the proliferation rates of MSCs by inducing apoptosis [13,14,15]. Xi. Chen et al. in 2014 demonstrated bacterial and viral stimulations of murine MSCs to express Toll-like receptors (TLR)3 and TLR4, respectively. Specific stimulations did not affect the self-renewal and apoptosis properties of MSCs with TLR3 but instead promoted their differentiation into osteoblasts, but the reverse of this observation was observed with TLR4 [13]. Reports from Hou et al., LPS were demonstrated to induce apoptosis in human umbilical cord mesenchymal stem cells via caspase activation. However, the authors concluded that pre-treating the MSCs with a low dose of LPS protects the cells by upregulation of FADD-like IL-1B- converting enzyme-inhibitory protein (c-FLIP) [14]. Hui Qie et al. in 2017 studied the effects of the MSC conditioned media on the human gingival fibroblasts exposed to LPS. MTT assay assessed cell proliferation, and apoptosis was assessed by flow cytometry. Their results showed significantly decreased cell proliferation with increased caspase activation and enhanced apoptosis when compared to cells exposed to the conditioned media [15].

By contrast, other studies have shown increased proliferation rates through NF-kB activation via TLR4 [16]. Earlier studies conducted on quantifying TLR4 and CD14 expression in adipose tissue-derived MSC in osteogenic differentiation have confirmed the expression of TLR4 but not CD14. The authors also demonstrated that LPS activated NF-kB via TLR4 and concluded that LPS induces proliferation and osteogenic differentiation through activation of TLR4 [16]. Furthermore, LPS alters the osteogenic potential of various types of MSCs [16,17].

Such conflicting results may be observed due to different concentrations, source, and purity of LPS used in other experiments. The lipid structure in the LPS from different sources differs, leading to a possible recognizable virulence, including their ability to activate specific TLR’s [18]. Furthermore, the heterogeneity in fatty acids is also reported to influence inflammatory response [18]. Reports indicating co-isolation of the components during the preparation of LPS may also be a reason for such reported ambiguity [19]. Christian Behm et al. al in 2020 aimed to “define the influence of structure versus purity of P. gingivalis LPS on the immune response of human PDLSCs and GMSCs”. Their methodology followed verification of surface MSC markers using flow cytometry, cell viability based on MTT assay, Quantitative PCR for gene expression, and ELISA for cytokine assay. The authors have shown increased responses of PDLSCs to more commonly used “standard” LPS from P. gingivalis compared to “ultrapure” LPS [19].

We have demonstrated previously that cells from the periodontal ligament of freshly extracted teeth can be isolated and characterized [20]. The isolated cells were earlier evaluated for cell proliferation assessed by manual cell counting and VEGF secretion [21]. Currently, there have been no reports that compare the response of 3D PDLSC’s and 2D PDLSC’s to LPS in terms of cell proliferation and or osteogenic differentiation and it is crucial to understand the effects of LPS in more physiologically relevant 3D cultures. Such studies may provide insights and help better understanding their biological and osteogenic abilities. Therefore, the objective of the present study was to compare the potential differential effects of LPS on the PDLSC’s 2D versus 3D culture model.

## 2. Materials and Methods

### 2.1. Cell Proliferation Studies

Previously isolated and characterized human periodontal ligament-derived mesenchymal stem cells [20] were used in this study. The cryopreserved PDLSCs were retrieved from the cryo-tank and expanded in a T75 flask with complete culture media (CCM) consisting of 88% Dulbecco’s Modified Eagle Medium (DMEM) (Gibco, New York, NY, USA), 10% fetal bovine serum (Gibco, New York, USA), 1% GlutaMAX^TM^ L-glutamine (Gibco, New York, NY, USA), 1% penicillin-streptomycin (Gibco, New York, NY, USA), ten ng/mL basic fibroblast growth factor (Gibco, New York, NY, USA). Upon 80% confluence, the attached cells were trypsinized (TrypLE express, GIBCO, New York, NY, USA), centrifuged, and resuspended in CCM, and passage (P)4 cells were plated in three 96 well plates at a density of 1.8 × 10^4^ cells per well in a 96 well plate. The cells were allowed to attach overnight.

Preparation of the LPS toxin: The LPS from *E. coli* (O111:B4) was commercially obtained (Sigma-Aldrich, Darmstadt, Germany). The powder was dissolved in 1 mL of sterile-filtered DMEM (Gibco, New York, NY, USA). The 1 mg/mL stock solution was further diluted using sterile PBS solution to prepare the final required concentrations of 5, 10, and 20 µgm/mL fresh every time.

LPS Treatment: following cell attachment, the CCM was aspirated, and the cells were treated with prepared LPS at 5, 10, and 20 µg/mL concentrations for 24, 48, and 72 h. All experiments were performed in triplicate.

MTT Assay: The MTT kit was obtained from Sigma, and the cell proliferation assay was carried out following the manufacturer’s recommendation. The PDLSCs were added with 20 µL of yellow MTT tetrazolium reagent and incubated for 4 h. Following this, the cells were observed under the microscope for the formazan crystals. On confirmation of the purple crystals, 100 µL of solubilization solution was added to each well and incubated overnight. This step was repeated for all three plates treated for 24, 48, and 72 h. The plates were read using an ELISA microplate reader (infinite 200) at 570 nm with a reference wavelength of 630 nm. All values recorded were tabulated for statistical analysis.

### 2.2. Osteogenic Induction

The P4 cells were plated at a density of 5 × 10^4^ in a 24 well plate. The cells were cultured in CCM until they reached 80% confluence. Upon confluence, we replaced the regular CCM with osteogenic media (StemPro osteogenesis kit, ThermoFisher Scientific, Waltham, MA, USA) with freshly prepared LPS solution at 5, 10, and 20 µg/mL concentrations. The osteogenic media with LPS was replaced every fourth day for a total of 28 days.

Alizarin-Red staining: At the end of Day 28, the cells were fixed using 4% formaldehyde. The fixed cells were washed in PBS to stain further using Alizarin-Red (MERCK, Darmstadt, Germany). The stained cells’ photomicrographs were captured using a 12-megapixel digital camera attached to a Nikon eclipse Ti-U inverted bright field microscope.

Extraction of Alizarin-Red stain for osteogenic quantification—the cetylpyridinium chloride (CPC) (MERCK, Darmstadt, Germany) extraction method was employed to extract the stain. Briefly, 350 mg CPC was dissolved in 10 mL of double-distilled water. We added an appropriate amount (2.3 mL) of the prepared CPC solution into each well, and the plate was incubated at 37 °C for 2 h with mild shaking. The plate was read using a microplate reader (infinite 200) at 405 nm wavelength. The absorbance readings were tabulated for statistical analysis.

### 2.3. Prearation of Clumps of Periodontal MSCs (C-MSCs)

This study used the method developed by Kittaka et al. to create clumps of MSCs [9]. Briefly, the P4 PDLSCs were seeded at a density of 7 × 10^4^ in 24-well plates and cultured with a growth medium supplemented with 50 µgm/mL of L-ascorbic acid (Sigma) for seven days. To obtain C-MSCs, confluent cells that had formed as a cellular sheet, consisting of the extracellular matrix produced by MSCs, were scratched with a micropipette tip and then were torn off. The MSC/ECM complex was then detached from the plate’s bottom in a rolled sheet shape and incubated for a day.

LPS treatment for the C-MSCs: The formed clumps of MSCs were carefully transferred into a 24-well ultra-low binding plate (Corning, New York, USA) and cultured in a regular culture medium added with different concentrations (5, 10 or 20 µgm/mL) of *E. coli* derived LPS for 3 and 5 days. The medium was replenished every third day. All experiments were performed in triplicate.

### 2.4. ATP-Based Cell Proliferation Assay for C-MSCs

An ATP-based kit (ATP lite 3D-Perkin Elmer, Waltham, MA, USA) was used in this experiment. After 3 (72 h) and 5 (120 h) days of LPS treatment, to 100 µL of culture volume, 100 µL of substrate solution was added, followed by shaking the plate for five minutes. The plate was incubated at room temperature (RT) for 20 min and then mixed vigorously by pipetting about 50 µL up and down. The mixed solution was then transferred (50 µL) to Optiplate. The plate was sealed, and the luminescence was read using a luminometer.

### 2.5. Osteogenic Induction of Clumps of MSCs

The clumps of cells were treated with osteoinductive medium (StemPro osteogenesis kit) (ThermoFisher, Waltham, MA, USA) with 5, 10, or 20 µgm/mL LPS for four weeks. C-MSCs treated with osteoinductive media (OIM) alone formed the positive control group, while C-MSCs treated with regular CCM formed the negative control group.

### 2.6. Raman Spectroscopic Analysis

The formaldehyde-fixed cell clump specimens were analyzed for their osteogenic potential using Raman spectroscopy. The device (Horbia iHR550, Horiba Jobin Yvon, Edison, NJ, USA) was set to a full scan range of 400–3200 cm^−1^ (785 nm diode laser as an excitation source power of around 20 mW at the sample for 30 s). Four accumulations were averaged for each spectral acquisition using 600 lines/mm grafting. A 50× objective was used to focus the light into a ~1 µm spot at 35 mW on the sample. The instrument was calibrated every fourth scan after the initial Si wafer is known to peak calibration at 520.7 cm^−1^. The RT was maintained at 24.5 °C and humidity at 53.5%. The surfaces at different sites were mapped through a 0.5 µm spacing with ten complete overlapping Gaussian lines for further calculations. For the centroid cluster measurements, the clumped cells’ mineral and organic bands were specifically examined to obtain the average mineral content within each region via three measurements. Principal component analysis (PCA) was used to analyze the spectral changes in the cell clumps. The Raman intensity peaks at 960 cm^−1^ (PO4-3) were proportional to the amount of calcium content. Bands at 785 cm^−1^ were assigned to the ring-breathing modes of DNA/RNA bases, such as adenine, cytosine, guanine, thymine, and uracil. The bands at 855 cm^−1^ were assigned for proline and hydroxyproline.

### 2.7. Multiplex Cytokine Immunoassay

To evaluate the inflammatory cytokines secreted under the influence of bacterial LPS, a comprehensive cytokine secretion profile of PDLSCs was analyzed using an antibody array recognizing IL-1β, IL-6, IL-8, and TNF-a cytokines. The cells from P5 were cultured and converted into clumps as previously described. The C-MSCs were kept under osteogenic induction to simulate the clinical osteogenic requirement during the tissue regeneration process. The osteogenic media was added with 0, 5, 10, or 20 µgm/mL different concentrations of LPS. The media with the toxin were changed every third day. The cells were under osteogenic induction for four weeks. The supernatants were collected and stored at −80 °C until further use. ProcartaPlex (Invitrogen, Waltham, MA, USA) multiplex immunoassay was employed to detect all four selected cytokines’ simultaneous detection and quantitation. This test uses differentially dyed capture beads for each target in a multiplex “ELISA-like” assay.

The procedure employed was as per the manufacturer’s instructions. Briefly, the frozen supernatant samples were thawed and centrifuged at 1000× *g* for 5–10 min. The wash buffer concentrate was diluted, and cortisol beads were prepared. A final volume of 250 µL of each antigen standard was prepared. Each different antigen standard set vials were centrifuged at 2000× *g* for 10 s. Then, 50 µL of sample-specific buffer was added to each vial. The previously collected supernatant was used to dissolve the standard. The vials were gently vortexed for 10 s and centrifuged at 2000× *g* for another 10 s. For complete reconstitution, they were incubated on ice for 10 min. Each vial’s entire contents were pooled into one vial, filled with a sample-specific buffer to a total volume of 250 µL. The vial was vortexed for 10 s and centrifuged at 2000× *g*.

A 4-four serial dilution of the reconstituted standards was prepared using the PCR tube strip. The tubes were labelled Std-1 to Std-8. Then, 200 µL of reconstituted antigen standard was added to the first tube and labelled as Std-1. Following that, 150 µL of sample type-specific standard buffer was added into Std Tubes 2–7. Furthermore, 50 µL reconstituted antigen was then transferred from Tube 1 to Tube 2 and mixed. Then, 50 µL of the mixed standard was transferred from Tube 2 to Tube 3 and mixed. Steps were repeated for Std Tubes 4–7. Then, 200 µL of culture media was added into Tube 8 that served as blank. The tubes were kept on ice until use.

Assay protocol—the plate map was defined and labelled for standards, samples, and blank. The magnetic bead solutions of 50 µL were added to the plates, and the magnetic beads were subsequently washed carefully. Wash buffer (150 µL) was added into each well and kept for 30 s to allow the beads to accumulate at the bottom of each well, followed by careful removal of the buffer. The 96-well flat-bottom plate was removed from the hand-held magnetic plate washer to add 50 µL of prepared standards or samples into dedicated wells. For wells designated as blanks, an additional 50 µL of cell culture media was further added. The plate was sealed and covered with a blank micropipette lid and was incubated overnight. Subsequently, the plate was shaken at 500 rpm for 30 min at RT and carefully washed on the following day. The detection antibody mixture was added into each well and incubated for 30 min at RT at 500 rpm after sealing. The plate was again washed. The 96-well plate was prepared for analysis by adding 120 µL of reading buffer into each well. The plate was sealed, covered with a microplate lid, and incubated for 5 min on a plate shaker at room temperature at 500 rpm and luminescence readings obtained using a microplate reader.

### 2.8. Statistical Analysis

Data were analyzed using the SPSS version 25. The values recorded were normalized and tabulated and presented as mean ± SD. ANOVA and post hoc Tukey tests were employed to test significance. All statistical analyses used a 95% confidence limit so that *p*-values of ≤0.05 are considered statistically significant.

## 3. Results

### 3.1. MTT Assay

The mean cell proliferation rates of the PDLSCs treated with LPS were compared to their controls using an ANOVA analysis. At 24 h, LPS increased the proliferation of the PDLSCs at all three concentrations (5, 10, and 20 µgm/mL) when compared to the untreated cells. However, this increase was not statistically significant. Interestingly, at 48 h and 72 h, LPS decreased the proliferation of PDLSCSs when compared to untreated cells. This decrease was statistically significant (*p* ≤ 0.01). The post hoc Tukey test revealed significant differences between all three concentrations (*p* ≤ 0.05) at 48 h except between 10 µgm/mL and 20 µgm/mL. However, at 72 h, significant differences between 0 µgm/mL and 5, 10, and 20 µgm/mL were observed but not between 5 µgm/mL 10, and 20 µgm/mL and between 10 µgm/mL and 20 µgm/mL. (Figure 1 depicts the proliferation rates of PDLSC’s with LPS treatment).

### 3.2. Alizarin Red Quantification

A decreasing amount of calcium production was observed when comparing the mean Alizarin-Red stain content following treatment of cells with different concentrations of LPS treated groups with the control group. The ANOVA analysis showed significant differences between the groups (*p*-value 0.005). The post hoc Tukey analysis showed significant differences between 0 µgm/mL and 10, and 20 µgm/mL (*p* ≤ 0.05) but not between 0 and 5 µgm/mL and ten and 20 µgm/mL (*p* > 0.05). Figure 2 shows the comparison of Alizarin-Red between different groups.

### 3.3. Creation of C-MSCs

The PDLSCs seeded at a density of 7 × 10^4^ in a 24-well plate and cultured in CCM supplemented with 50 μgm/mL L-ascorbic acid for one week. Gentle scratching of the confluent cells using a micropipette tip resulted in a cellular sheet-like structure detached from the well’s base. The sheet could be easily rolled up for further incubation successfully, resulting in cells clumping themselves along with the production of extracellular matrix forming the C-MSCs (Figure 3).

### 3.4. ATP-Based Cell Proliferation Assay

When comparing the mean ATP production rates of the C-MSCs treated with LPS, no significant differences were observed between the groups at 72 h and 120 h (*p*-value 0.790 and 0.063, respectively). Accordingly, the post hoc Tukey test revealed no significant differences between the groups. However, when we compared the four different concentrations, a gradual reduction in ATP production was observed in a dose-dependent manner, but this was not statistically significant. (Figure 4 shows the comparison of ATP production between different groups).

### 3.5. Quantification of Osteogenic Differentiation of C-MSCs

In the Raman analysis, the spectroscopic region representing the characteristic bands of minerals is most intense for hydroxyapatite (calcium formation), showing symmetric phosphate stretching (A1) be labelled as PO_4_^3−^; v1. It was assigned at 960 cm^−1^. The band was observed to be extremely sensitive to the presence of calcium minerals. The assigned bands were observed at the highest intensity in specimens treated with osteogenic medium alone; however, the samples treated with LPS showed low-intensity peaks concerning calcium mineral (Figure 5A). Several variations were observed at 785 cm^−1^ (C-O and C-C vibrations and all components of DNA nucleic bases), representing DNA/RNA bases. The groups’ peak intensity showed changes amongst the specimens treated with OIM and those treated with LPS (Figure 5B). The peaks at 785 cm^−1^ were assigned to the ring breathing modes of DNA/RNA bases, such as adenine, cytosine, guanine, thymine, and uracil (Figure 4B). The difference between these nucleic acid-associated peaks was mainly due to the difference in LPS concentrations. In Figure 5C, it was reasonable to assume the configuration of hydroxyproline within the specimens.

Interestingly, the relative intensity of the hydroxyproline/proline bands, normalized at 850–860 cm^−1^, increased in OIM samples, indicating cell proliferative activity (C-C proline, hydroxyproline). The CH_2_ wag is considered a protein component and was chosen to further analyze the general matrix’s protein activity within the C-MSCs. The peaks at 785 cm^−1^ and 855 cm^−1^ were lower than the group treated with OIM alone. Figure 5D indicates the calcium phosphate peak intensities after normalization. The display components are ascribed to mineralization with broad detectable peaks at 960 cm^−1^. There is a significant peak for samples treated with 5 µgm/mL LPS.

### 3.6. Multiplex Immunoassay

The selected cytokine secretion profile appeared to favor pro-inflammatory mediators. The ELISA multiplex assay confirmed that LPS treatment resulted in increased secretion of IL-6 (up to 20-fold rise) and IL-8 (up to 40-fold rise) on Day 3 compared to the control group, and this observation was statistically significant. When the cells were continuously treated with LPS, the secretion profile of IL-8 showed a downregulation on Days 6, 12, and 24. IL-6 levels were higher even during the later days, like Days 6, 12, and 24, and the differences between the groups were significant until Day12 but not Day 24. A progressive increase in IL-6 levels was observed even within the control group. The IL-8 levels were significantly less through Days 6, 12, and 24 when compared to Day 3. Upon LPS stimulation, the C-MSCs cells did not secrete TNF-a and IL-1β even on Day 3 through to Day 24. Figure 6 depicts the comparison of IL-6 and Il-8 secretion by C-MSCs under osteogenic induction with or without LPS at different time intervals.

## 4. Discussion

The multiplicity of gram-negative organisms and their LPS rich cell walls are relevant and widely recognized in periodontal disease development and progression. Such endotoxins can induce significant responses in a variety of cell types, including PDLSCs. In the current in vitro study, we evaluated the effects of LPS on proliferation, osteogenic abilities, and cytokine profiles of the 2D and 3D PDLSC model. The effects of LPS on various stem cells or related cells, including the PDSLCs, have been investigated by many researchers; however, the documented results are extremely conflicting. Yamasaki et al. in 1998 demonstrated decreased cell proliferation in periapical tissue-derived fibroblasts when treated with sonicated gram-negative bacterial extracts [22]. Herzmann et al., in 2017, showed LPS stimulates proliferation and osteogenic differentiation via activation of TLR-4 receptors [16]. However, Lertchirakarn et al. in 2017 studied the effects of LPS on cell proliferation and osteogenic differentiation of stem cells from the apical papilla (SCAPS) [23]. Their observations indicated that LPS does not have any significant effect on both cell proliferation and osteogenic potential. However, the authors concluded that at high concentrations (>5 µgm/mL), LPS could significantly enhance the expression of bone sialoprotein (BSP) expression. Kooshki et al. in 2018 preconditioned Bone marrow MSCs with LPS on modified Poly-L lactic acid nanofibers and studied their proliferation and osteogenic potential. Their results pointed towards LPS ameliorating the proliferation and osteogenic potential of LPS pre-treated bone marrow MSCs [17]. Similarly, other researchers have demonstrated LPS to decline the proliferative ability of various MSC cell types. In contrast, others have reported the stimulatory effects of LPS on cell proliferation. The varied observations reported could be due to non-standardized concentrations of LPS being used in the experiments. These conflicting reports led us to hypothesize that LPS would induce differential effects on PDLSCs cultured in may exert in 2D versus 3D culture models.

In 2016, Andrukhov et al. compared the response of PDLSCs to P. gingivalis LPS, and *E. coli* derived LPS. They reported no significant differences between the two sources of LPS in the absence of CD14 [24]. Our study used LPS derived from E. coli, and at 24 h, all concentrations used in the 2D culture model enhanced cell proliferation, but this observation was not statistically significant. This increase can be attributed to the initial protective response of PDLSCs to inflammatory insults of LPS. However, at 48 and 72 h, the LPS significantly decreased the proliferation of PDLSCs. The reduced cell proliferation observed in our study may be due to the direct induction of apoptosis by LPS through the triggering of caspases. LPS induced apoptosis is reported to mediate through the classical NF-kB pathway signalling through the Toll-like receptor 4 (TLR) present on the MSCs [25], including PDLSCs. These TLR-4 receptors recognize LPS because of their specific molecular pattern [13]. It is also reported that TLR-4 regulates cell proliferation in MSCs in the presence of LPS. It is reported that soluble CD14 enhances the response of PDSLCs when exposed to LPS [24]. The authors demonstrated decreased levels of CD14 in PDLSCs, indicating a lower response. This report commands an additional study on evaluating CD14 in our established PDSLC cell line. We selected 5, 10, and 20 µgm/mL concentrations in our study since many researchers have already demonstrated that LPS at low concentrations is cytoprotective. Yu Sen Hou et al. demonstrated human umbilical cord-derived MSCs undergo apoptosis when treated with LPS, but pre-treating with low dose LPS interestingly prevents apoptotic cell death [14]. This observation indicates that low-dose LPS can offer a cytoprotective role. It is reported that pre-treating MSCs can benefit cells by conditioning them to produce specific extracellular vesicles that can polarise macrophages towards the anti-inflammatory M2 phenotype [26]. However, our study’s selected concentrations are higher, thereby exterminating the chance of the reported cytoprotective role offered by LPS. On post hoc Tukey evaluation, at 72 h, there was no significant difference between 5, 10, or 20 µgm/mL LPS levels; however, at 48 h, there were significant differences between 5 and 10 µgm/mL concentrations but not between 10 and 20 µgm/mL. This result infers that LPS at 5 µgm/mL is potent enough to induce significant TLR-4 mediated apoptosis levels. This observation is in line with the study done by Hou et al. [14]. however, the MSC cell type employed varies compared to our study.

Interestingly, when the 3D C-MSCs were treated with different concentrations of LPS, no significant difference in ATP production between the LPS treated groups and the control group was observed. The C-MSCs produced by curling a confluent monolayer supplemented with a growth medium with L-Ascorbic acid produce extracellular complexes (MSC/ECM, cellular complexes) [9,27]. Indeed, it has been reported that ascorbic acid supplementation to the growing cells results in the production of abundant amounts of collagen and glycosaminoglycans [28]. Moreover, H Cohly et al., in their study on culture conditions affecting LPS inducibility in inflammatory conditions, showed that the environmental factors in the culture media alone could alter the biological nature of the cells [29]. Following these reports, we assume that the production of MSC/ECM complexes by C-MSCs has a protective attenuating role on the cytotoxic effects of LPS.

When the monolayer PDLSCs were cultured under the influence of OIM with LPS for four-weeks, a significant reduction in the Alizarin-stained calcium content was observed in groups treated with ten and 20 μgm/mL LPS when compared to 5 μgm LPS and the control group as detected by CPC extraction method. This decrease is following many previously published studies on other types of MSCs. We believe this could be indirectly attributed to the reduced cell numbers that we reported due to the cytotoxic effects of LPS.

Short et al. employed Raman analysis to detect the spectra of proliferating and nonproliferating cells in culture plates [30]. Their observations confirmed that the process could monitor biochemical changes in lipids, proteins, DNA, and RNA due to cell proliferation. Omberg et al. used the factor analysis of Raman spectra to differentiate tumorigenic and non-tumorigenic cell lines [31]. Similarly, many other researchers [32,33] have studied the Raman spectra of various neoplasia, epithelial cancers, and pre-cancers. These studies demonstrated Raman spectroscopy’s applicability to identify the differences observed between normal and abnormal states and cell cultures. However, as a limitation, these are usually only a qualitative understanding of the underlying biochemical changes. No study using Raman has been employed to analyze the biochemical changes in a three dimensional cell culture model to date. In our observations, the bands that we obtained from different C-MSC specimens were acquired in the region of 400 cm^−1^ and 3200 cm^−1^. The principal bands identified were the P–O bond at 960 cm^−1^ of the mineral component. This spectroscopic region represents the characteristic band of minerals that are most intense for calcium formation. This observation follows a study conducted by Smith et al. and McManus et al. They employed Raman spectroscopy to detect early osteogenic commitment in primary cells [34] and monitoring osteogenic cells, respectively [35]. Similarly, Mitchell et al., in 2015, demonstrated the spectral changes associated with adipogenesis in MSCs [36].

In our study, the intensity of the calcium peaks showed highest amongst those in the osteogenic media treated group than the other groups with minor changes in between those groups. The peaks at 785 cm^−1^ assigned to the ring-breathing modes of DNA/RNA bases, such as adenine, cytosine, guanine, thymine, and uracil, represent DNA content. The difference between these nucleic acid-associated peaks was mainly due to the addition of different cell counterparts. Notably, Raman spectra of the nuclei display pronounced peaks at 496, 532, 595, 667, 682, 729, 783, 1013, 1062, 1096, 1178, 1208, 1243, 1334, 1371, 1419, 1484, 1505, 1531, 1576, 1663, and 1690 cm^−1^ that match the Raman spectrum of pure DNA. The vibrations at 855 cm^−1^ assigned to proline and hydroxyproline indicate the cell proliferative activity that translates to collagen-related protein production. This observation confirms the presence of ECM complexes within the C-MSCs. However, experiments conducted using Raman spectra provided qualitative results, and further quantitative studies in this direction are recommended. This qualitative observation is one of the limitations of our study.

Soluble proteins secreted by the cells in response to infection or a toxin having low molecular weight glycoprotein are called cytokines [37]. They are mediators of cell function and are produced by a variety of cell types. Cytokine profiling is a potent way to understand the cellular defence against a pathogen [38]. To date, many researchers have reported the levels of different cytokines in patients with periodontitis. Zohaib et al., in 2016, conducted a systematic review and meta-analysis of papers ranging from 1977 up to 2016 on the cytokine profile in chronic periodontitis patients with and without obesity [37]. Their observations concluded the level of local inflammation has a greater influence on pro-inflammatory markers than systemic problems. Recently, in 2020, the cytokine profiles of healthy and diseased sites in periodontitis patients was reported and indicated that there were increased levels of IL-8 and MIP-1α in healthy sites and disease sites showing high levels of Th-17 related cytokines and TGF-β [39].

Any cytokine closely associated with an inflammatory response’s onset and progression is identified as an inflammatory cytokine. The most established cytokines in periodontitis patients in this regard are IL-1α, IL-β, IL-6, IL-8, and TNF-α. Exclusive attention has been paid to these cytokines over the years due to their ability to increase bone resorption [40]. In contrast, during the early osseous repair, IL-6, 8, and TNF-α are detected to recruit MSCs to induce bone formation [41]. In bone marrow-derived MSCs, IL-6 has been shown to stimulate osteogenesis and supportively influence the mitogen-activated protein kinase signalling cascade required for bone formation [42]. The effect of various pro and anti-inflammatory cytokines on osteogenic differentiation of MSCs has been controversial [43]. Hence, we opted to include these four cytokines in our study.

The multiplex ELISA demonstrated increased secretion of IL-6 and IL-8 in the early part of the experiment. This observation is following studies reported in the literature [44,45], where the authors have shown elevated levels of IL-6 and IL-8 within the first 24–72 h after bone injury. However, when the cells were continuously treated with LPS, the secretion profile showed down-regulation on Days 6, 12, and 24 for IL-8 but not IL-6. This observation may be correlated with the increased levels of IL-8 observed at healthy sites in a study conducted by Miranda et al. [39]. IL-8, also known as CXCL8, is a neutrophil-specific chemokine. Therefore, it is hypothesized that IL-8 may play an innate immunity role by expressing itself more in disease-free states than in persistent inflammatory states. This role may be a reason for the decreased IL-8 levels over time because of the reported immunomodulatory properties of MSCs. Angela et al. in 2016 reported an initial secretion of pro-inflammatory cytokines within the injured site followed by a shift to anti-inflammatory cytokines [43]. However, a conflicting report of IL-8 being a biomarker for longevity at inflammatory sites for weeks has been published [46]. Further studies specifically on IL-8 in different healthy and or inflamed conditions may help understand better. IL-6 levels, on the other hand, remained high through Days 6, 12, and 24. However, the levels between all groups were significant till Day 12 but not 24. Interestingly, the levels of IL-6 increased even in the control group. This increase may be attributed to the osteogenic influence itself following the reports by Xie et al., who demonstrated increased IL-6 levels during the osteogenic induction of bone marrow-derived MSCs [47].

Interestingly, C-MSCs exposure to LPS did not secrete any detectable IL-1β and TNF-α levels over the entire time. The IL-1 family of eleven cytokines is controlled on transcriptional and posttranscriptional levels [48]. This regulation may be due to its innate inflammasome [49] dependency. LPS may induce the production of an inactive form of IL-1β called pro-IL-1β; however, it needs to be processed by caspase-1 [50] by activation of inflammasome molecules. Based on our observations, we hypothesize that LPS treatment of C-MSCs may not activate caspase-1 to cleave intracellularly produced proIL-1β. Further studies targeting the pathways involved in its processing are needed. On the other hand, TNF-a has been reported as a cytokine executed through specific TNFR family members. Lieke et al. examined TLR function on various MSCs and found that stimulation of these cells resulted in increased secretion of limited cytokines. Interestingly, TNF was undetectable after TLR stimulation. Their experiments indicated that it was due to the TNF promoter. The authors confirmed a similar observation on bone marrow-derived MSCs even after LPS stimulation. They reported that these cells can respond to TNF [51] stimulation but cannot produce TNF even after LPS insult. Our observations on TNF-a follows this report. TNF-α was undetectable in our samples. Hence, we hypothesize the presence of inactivated TNF promoter even in PDLSCs.

Since inflammation is crucial in the pathogenesis of periodontal disease, controlling inflammation in all possible ways is beneficial to treating periodontal disease [52]. Previous studies incorporating stem cell-based treatment strategies have demonstrated success in various capacities. The neutrophil recruiting capabilities of IL-8 that may further destroy the periodontal tissues by releasing metalloproteinases are well documented in the literature [53]. Similarly, IL-1β and TNF-α synergistically increase IL-6 several times more potent than LPS itself [54]. Based on our observations, the PDLSCs do not secrete two out of four threatening cytokines. This observation allows us to assume reduced activation of NF-kB and mitogen-activated protein kinase (MAPK) signalling pathways.

The abilities of stem cell-based therapies are chiefly based on their ability to withstand inflammatory insults. In the present work, we demonstrated the 3D PDLSC’s potential to withstand better the effects of LPS induced inflammation and its attenuating effects on the cell’s osteogenic potential. Hence, the transplantation of such 3D PDLSCs with or without the use of scaffolds to periodontal defects may hold a promising therapeutic potential. Furthermore, previous studies by Kittaka et al. have shown that such bone marrow-derived 3D cultured cells can be transplanted in small and relatively large, deep defects such as bone fractions, palatoschisis, and periodontal defects [9,10]. However, larger bone defect cases may not be addressable by C-MSC’s, which are usually not the case in periodontal disease. Further, the 3D culture model can regulate osteogenic differentiation significantly better than the traditional 2D culture for more effective periodontal regenerative therapies.

Potential limitations of the current work include the lack of investigation on NF-kB translocation upon stimulating the PDLSCs with TLR agonists. Research in this direction may shed light upon the exact reason for the lack of TNF-α secretion by the PDLSCs even after LPS stimulation. Secondly, the purity of the LPS employed can also be a limitation since the purity of LPS is reported to have a profound effect on the PDLSCs. Finally, even though the 3D MSC’s can mimic the in vivo physiology, an animal model that investigates by addressing the limitations mentioned above would help understand the molecular mechanisms and the potential interactions involved.

## 5. Conclusions

In the current study, we showed that in the presence of LPS PDLSCs proliferate at a slower rate and their osteogenic abilities are impaired. Our data indicate that C-MSCs are more resistant to the effects of LPS and retain the osteogenic potential than cells grown in monolayer cultures, with increased IL-6 secretion suggesting a positive influence on osteogenic induction. Furthermore, our observations support that proIL-1β produced by PDLSCs is not cleaved by caspase-1, and PDLSCs have an inactivated TNF promoter. Such decreased cytokine response and immunosuppressive properties of MSCs make them a perfect model for personalized cell-based therapies. The 3D clumps have the potential to be employed in scaffold-free periodontal regenerative treatment strategies.

## Figures and Tables

**Figure 1 jpm-11-00528-f001:**
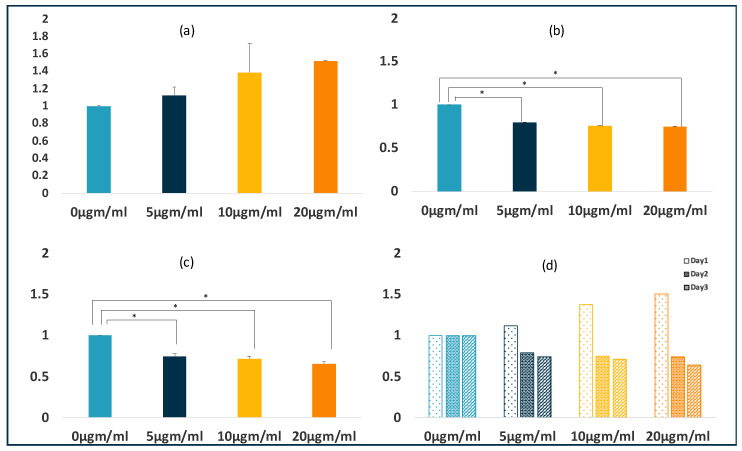
Effects of LPS on proliferation of PDLSCs in 2D culture. LPS inhibited the proliferation of PDLSCs. The graphs depicts the PDLSC’s proliferation after 24 (**a**), 48 (**b**), and 72 (**c**) hours of LPS treatment. Graph (**d**) represents the mean proliferation values of the PDSLC’s after 24, 48, and 72 h of LPS treatment. The values of three independent experiments are presented as means ± SD. * *p* ≤ 0.05.

**Figure 2 jpm-11-00528-f002:**
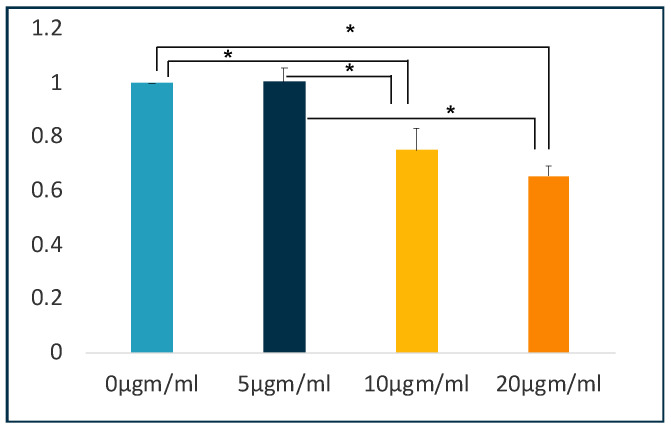
Quantification of Alizarin-Red. LPS affected the osteogenic capabilities of PDLSCs. The cells were cultured under osteogenic medium for four weeks in the presence of different concentrations of LPS. * *p* ≤ 0.05.

**Figure 3 jpm-11-00528-f003:**
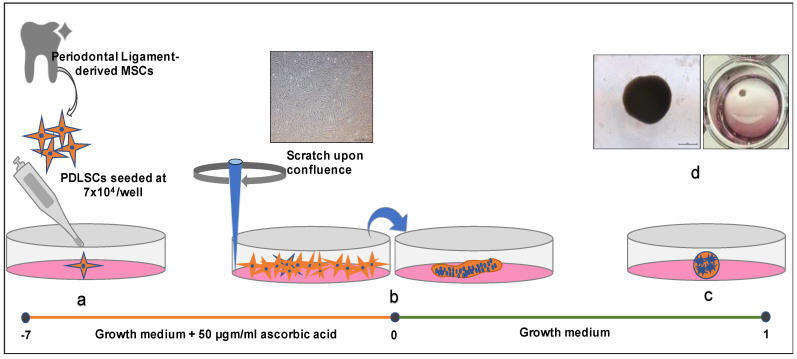
Preparation of C-MSCs. (**a**) PDSLCs were cultured in a 24-well plate with the addition of 50 µgm/mL L-ascorbic acid for seven days. (**b**) The confluent cells were scratched to gently tear off as a cellular sheet that consists of ECM. (**c**) After one-day incubation with growth media alone, cells rolled up to make spherical clumps. (**d**) C-MSC in a 24-well plate observed under a bright-field microscope. Scale bar 50 µm.

**Figure 4 jpm-11-00528-f004:**
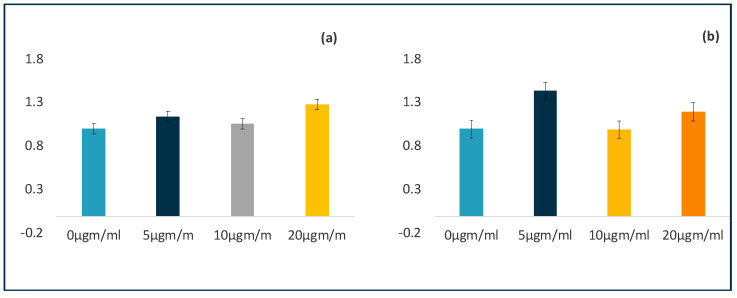
Effects of LPS on proliferation of PDLSCs in 3D culture. Clumps resist the effects of LPS when compared to cells grown in 2D. Graphical representation of comparison of ATP production by 3D clumps after 72 h (**a**) and 120 h (**b**) of LPS treatment. The values are of three independent experiments presented as means ± SD. * *p* ≤ 0.05.

**Figure 5 jpm-11-00528-f005:**
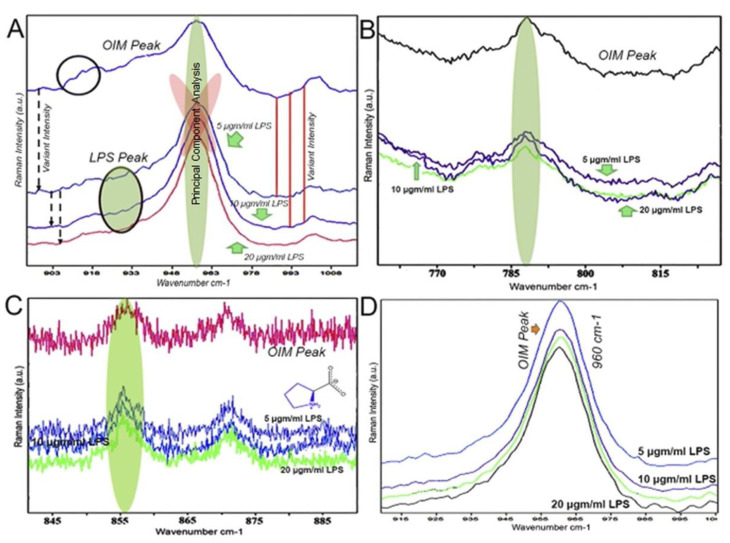
Observations under Raman analysis. (**A**) The peaks observed due to PCA at 960 cm^−1^ representing calcium were highest for C-MSCs under OIM alone (OIM/DF), and other peaks representing 5, 10 and 20 μgm/mL demonstrated qualitatively reduced intensity. (**B**) Waves assigned at 785 cm^−1^ representing DNA/RNA and (**C**) assigned at 855 cm^−1^ for proline and hydroxyproline showed relatively reduced peaks for C-MSCs compared to cells treated with OIM alone. (**D**) Indicates the calcium phosphate peak intensities with baseline correction of each spectrum with the Polyfit method before performing the normalization (scaling the data by calculating the average of the spectrum, standard deviation and subtracting each value of the spectrum by the mean: finally divided by the standard deviation).

**Figure 6 jpm-11-00528-f006:**
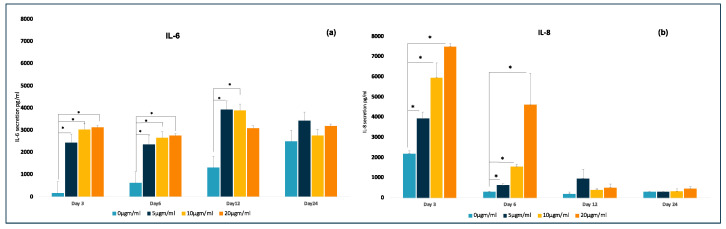
Effects of LPS on cytokine secretion profile of 3D PDLSCS. Cells were cultured under osteogenic induction and with LPS at different concentrations for different time intervals. Graphical representation of (**a**) IL-6 and (**b**) IL-8 secretion by 3D PDLSCs. The values of three independent experiments are presented as means ± SD. * *p* ≤ 0.05.

## Data Availability

Not applicable.

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
