# Peer review of "3D Clumps/Extracellular Matrix Complexes of Periodontal Ligament Stem Cells Ameliorate the Attenuating Effects of LPS on Proliferation and Osteogenic Potential"

_jpm, 2021, doi:10.3390/jpm11060528_

Round 1

Reviewer 1 Report

the article has a mailing list. The description of the research was made in a transparent manner. In the text of the policy polymic with the current scientific reports, which is its advantage. The authors used a variety of verification methods. The results are transparent. The discussion covers many aspects of the case. Of course, it would be possible to extend the description in places, but in my opinion the article may be in print. 

Reviewer 2 Report

I consider the work of great interest to readers and the experiments are intersting however I consider that a more reader-friendly version could ve more appropriate.There are is a lot of data and an accurate interpretation of results, however I would suggest the authors to take into consideration to coroborate their results with the clinical aspects: how does their study influence the knowledge that we previously had and in short how do the results influence current periodontal therapy?

Reviewer 3 Report

The manuscript submitted to JPM entitled “3D Clumps of Periodontal Ligament Stem Cells Ameliorate the Attenuating Effects of Lipopolysaccharide” is an original article which aim to investigate the effect of lipopolysaccharide on periodontal-ligament-derived mesenchymal stem cells in monolayer and 3D culture.

On my opinion the article is interesting, well written, with good English.

However, I highlighted some issues.

  • Title: Please improve the title with more details on your study.
  • English language. Minor spell check is required.
  • Please structure the abstract to attract the reader's attention and adapt it accordingly.
  • Please improve this section. Specify if there are other similar studies. Better specify the objectives and methods of the study. I suggest inserting the following sentence and reference at line 62 on page 2: << The periodontal ligament (PDL) represents a fibrous network connecting the cementum of the tooth root and the alveolar bone. It serves many functions, such as tooth support, nutrition, and protection [https://doi.org/10.1177/0963689718807680]>>.
  • This section has been properly prepared.
  • Are there other similar studies that have shown similar results? Did the authors find limitations in their study by comparing it with other in the literature?
  • Materials and Methods: This section has been properly prepared.
  • Improve the quality and arrangement of figures.

Summary of abbreviations required at the end of the manuscript prior to “Reference” section.

After making the indicated changes, I am available for a second round of peer review.

Thanks for the opportunity to review this manuscript.

Round 2

Reviewer 3 Report

After the changes made, the article is suitable for publication.